# Independent Signaling of Hepatoma Derived Growth Factor and Tumor Necrosis Factor-Alpha in Human Gastric Cancer Organoids Infected by *Helicobacter pylori*

**DOI:** 10.3390/ijms24076567

**Published:** 2023-03-31

**Authors:** Kenly Wuputra, Chia-Chen Ku, Jia-Bin Pan, Chung-Jung Liu, Kohsuke Kato, Ying-Chu Lin, Yi-Chang Liu, Chang-Shen Lin, Michael Hsiao, Ming-Hong Tai, Inn-Wen Chong, Huang-Ming Hu, Chao-Hung Kuo, Deng-Chyang Wu, Kazunari K. Yokoyama

**Affiliations:** 1Graduate Institute of Medicine, College of Medicine, Kaohsiung Medical University, Kaohsiung 80708, Taiwan; kenlywu@hotmail.com (K.W.); r991046@gap.kmu.edu.tw (C.-C.K.); r060139@gap.kmu.edu.tw (J.-B.P.); csl@kmu.edu.tw (C.-S.L.); 2Regenerative Medicine and Cell Therapy Research Center, Kaohsiung Medical University, Kaohsiung 80708, Taiwan; pinkporkkimo@yahoo.com.tw; 3Cell Therapy and Research Center, Kaohsiung Medical University Hospital, Kaohsiung 80756, Taiwan; ycliu@kmu.edu.tw; 4Division of Gastroenterology, Department of Internal Medicine, Kaohsiung Medical University Hospital, Kaohsiung 80756, Taiwan; kjh88kmu@gmail.com; 5Department of Infection Biology, Graduate School of Comprehensive Human Sciences, The University of Tsukuba, Tsukuba 305-8577, Japan; kkato@md.tsukuba.ac.jp; 6School of Dentistry, Kaohsiung Medical University, Kaohsiung 80756, Taiwan; yclin@cc.kmu.edu.tw; 7Genome Research Center, Academia Sinica, Nangan, Taipei 11529, Taiwan; mhsiao@gate.sinica.edu.tw; 8Institute of Biomedical Sciences, National Sun Yat-sen University, Kaohsiung 80424, Taiwan; minghongtai@gmail.com; 9Division of Pulmonary and Critical Care Medicine, Kaohsiung Medical University Hospital, Kaohsiung 80756, Taiwan; 10Department of Internal Medicine, Kaohsiung Municipal Ta-Tung Hospital, Kaohsiung 80145, Taiwan; 11Department of Internal Medicine, Kaohsiung Municipal Siaogang Hospital, Kaohsiung 81267, Taiwan

**Keywords:** *Helicobacter pylori*, hepatoma-derived growth factor, invasion activity, gastric cancer organoids, tumor necrosis factor α

## Abstract

We prepared three-dimensional (3-D) organoids of human stomach cancers and examined the correlation between the tumorigenicity and cytotoxicity of *Helicobacter pylori* (*H. pylori*). In addition, the effects of hepatoma-derived growth factor (HDGF) and tumor necrosis factor (TNFα) on the growth and invasion activity of *H. pylori*-infected gastric cancer organoids were examined. Cytotoxin-associated gene A (CagA)-green fluorescence protein (GFP)-labeled *H. pylori* was used to trace the infection in gastric organoids. The cytotoxicity of Cag encoded toxins from different species of *H. pylori* did not affect the proliferation of each *H. pylori*-infected cancer organoid. To clarify the role of HDGF and TNFα secreted from *H. pylori*-infected cancer organoids, we prepared recombinant HDGF and TNFα and measured the cytotoxicity and invasion of gastric cancer organoids. HDGF controlled the growth of each organoid in a species-specific manner of *H. pylori*, but TNFα decreased the cell viability in *H. pylori*-infected cancer organoids. Furthermore, HDGF controlled the invasion activity of *H. pylori*-infected cancer organoid in a species-dependent manner. However, TNFα decreased the invasion activities of most organoids. We found different signaling of cytotoxicity and invasion of human gastric organoids in response to HDGF and TNFα during infection by *H. pylori*. Recombinant HDGF and TNFα inhibited the development and invasion of *H. pylori*-infected gastric cancer differently. Thus, we propose that HDGF and TNFα are independent signals for development of *H. pylori*-infected gastric cancer. The signaling of growth factors in 3-D organoid culture systems is different from those in two-dimensional cancer cells.

## 1. Introduction

Approximately 1.1 million patients were diagnosed with gastric cancer worldwide in 2020. Gastric cancer is among the three most common cancers in 23 countries and is regarded as the most common cancer globally [1]. The treatment of gastric cancer often presents a problem because patients are frequently diagnosed in the later stages of the disease with distant metastasis and chemoresistance [2]. In addition, the five-year survival rate for gastric cancer patients is about 30% [3]. Thus, there is a pressing need to understand the mechanisms of tumorigenesis of gastric cancer to improve diagnostic, prognostic, and therapeutic development. *Helicobacter pylori* (*H. pylori*) infection is known as the most common risk factor for gastric cancer, and which infected more than half of the global population [4]. *H. pylori* infection is associated with chronic gastritis, mucosa-associated lymphoma, and peptic ulcer. Chronic infection leads to the gradual progression of gastric atrophy, dysplasia, intestinal metaplasia, and finally to adenocarcinoma based on Correa’s axis of tumorigenesis [5]. In addition, *H. pylori* infection leads to the emergence of a tumorigenic subpopulation of cells that can survive and coexist with a DNA damage [6]. These abnormal tumorigenic gastric epithelial cells progress to gastric cancer [7].

Organoids prepared from gastric cancer tissue, normal organoids from the cancerous parts, and organoids from induced pluripotent stem cells (iPSCs) were established over a decade ago [8,9,10,11]. Even then, many questions remain unaddressed regarding the extent to which these cultures recapitulate normal gastric development and the mechanism of cancer progression infected by *H. pylori.* Here we focused on the growth factor, hepatoma derived growth factor (HDGF), and the key inflammation factor, tumor necrosis factor (TNFα). In earlier studies, we found that HDGF-deficient mice significantly inhibited *H. pylori*-induced neutrophil infiltration and reduced TNFα/Cyclooxygenase 2 (COX2) signaling leading to inflammation, thereby reducing tissue damage in the stomach. In time course experiments, TNFα acts as a priming signal to increase *H. pylori*-induced TNFα/COX-2/Prostaglandin F2 (PGF2) expression. Taken together, these data suggest that the nuclear factor kappa B (NF-*k*B) signaling plays a crucial role in inflammation [12]. To confirm this hypothesis, we examined the role of TNFα and HDGF using gastric organoids infected with *H. pylori.* This experiment is beneficial for the elucidation of the mechanistic role of *H. pylori*-infected gastric cancer development. It has been reported that HDGF upregulation by *H. pylori* infection can promote tissue damage and even cancer development by production of excess reactive oxygen species [13,14]. Previously, we reported that HGDF and TNFα were crucial for cancer initiation and development using two-dimensional (2-D)-gastric cancer cell lines [12]. However, the validity of these signaling pathways in the three-dimensional (3-D)-organoid gastric model system remains unknown. In this study, we found different signaling for cancer progression and invasion of human gastric organoids in response to recombinant HDGF and TNFα during the infection with *H. pylori*. TNFα is defined as the upstream partner of HDGF in gastric cancer cells in 2-D culture [12]; however, in (3D) organoids in culture, HDGF and TNFα are involved in independent signals in the development of *H. pylori*-infected gastric cancer. We also discuss the difference in growth factor signaling in the 3-D organoid culture system compared with 2-D cancer cell models [15,16,17].

## 2. Results

### 2.1. Injection of GFP-H. pylori into Gastric Cancer Organoid

To establish long-term infection with *H. pylori*, we generated human gastric cancer-derived organoids from the adenocarcinoma from gastric cancer regions including antrum and fundus by following the standard methods as described elsewhere [18]. We infected these organoids with various cytotoxin-associated gene (Cag) mutant *H. pylori* strains, before injection into immunodeficient mice as a cancer model. To monitor infection with CagA-green fluorescence protein (GFP)-labeled *H. pylori* Hp0547, we infected organoids and cultured them for 24 h at a multiplicity of infection (MOI) of 200 (Figure 1A). Injection efficiency and validity were confirmed by a microscope using GFP expressed by *H. pylori*. The fluorescence microscope showed that bacteria were indeed engaged in every intimate contact with the central cavity (Figure 1B,C). Previously, Bartfeld et al. [19] reported that a 2 h infection increased the expression of genes of mucus pit cell marker MUSC5AC, gland mucus cell marker MUC6, IL-8, NF-*k*B p65, CCL20, ICAM1, and gastric cancer marker-like chorionic gonadotropin beta. Here, we found that the expression of IL-8, NF-*k*B p65 subunit, CCL20, ICAM1, and beta-catenin as cancer stem cell markers was significant after infection for 24 h of infection, as was observed in the 2-D gastric cancer cells. We then examined the signaling of GFP within the organoids after cultivation for 1 week. Infections were continued for 1 week to monitor the tracing of the *H. pylori* in human stomach adenocarcinoma organoids (Figure 1D). There was no significant difference in the intensity of the injected GFP-*H. pylori* after one week cultivation, indicating that the *H. pylori* can possibly infect into the gastric organoids after microinjection into the cavity.

### 2.2. H. pylori Species with Three Different Toxicities Were Compared with the In Vitro Expansion of Gastric Cancer Organoids

*H. pylori* CagL Y58/E59, amino acid polymorphisms, can exploit higher expression of integrin alpha5 beta1 and gastritis in the upper stomach, and are associated with a 4.6-fold increase in gastric cancer risk [18]. *H. pylori* CagL-Y58/E59 isolates display greater inflammation in corpus and integrin alpha5 beta 1 expression in the upper stomach, where they commonly exit with the chief cells and output secretory cells in the mucosa. *H. pylori* infection-induced acid secretion in the corpus was reduced in human gastric corpus, suggesting that corpus inflammation and atrophy are two independent factors for hypoacidity in the stomach [20]. Moreover, hypochlorhydrias have been found to increase gastritis and then lead to the development of precancerous changes progressing into dysplasia or even gastric integrin [21].

The selected *H. pylori* gastric cancer strain was classified into three members according to their virulence. The strain with the highest CagA product toxicity was designated as 49503 (Hp60190 (ATCC 49503, CagA^+^) and the noncytotoxic strain as a mutant of the CagL gene, HpCagLm Y58D/59K, HpHtrA171/nonL and non-CagL Y58/E59 strains with moderate toxicity were generated and cultured based on the recommended protocol [18,22].

The growth of organoids was examined by bromodeoxyuridine incorporation assay and cytotoxic assays [18,22]. The cytotoxic activity of HCM-BROD-0045-C16 cancer model metastatic adenocarcinoma of stomach was designated as 1.0 and compared with organoids infected by each *H. pylori* ATCC 49503, Hp1033, HpCagLm Y58D/E59K, and HpHtrA171/nonL and non-CagL Y58/E59. The cytotoxic activity in each case did not change at all (Figure 2A,B). The morphology of organoids with or without *H. pylori* infection is shown in the left panels. In addition, we detected that the NF-*k*B p65 subunit was translocated into the nuclei after incubation with *H. pylori.* It has been reported that the expression of IL-8 chemokine attracted neutrophils and thereby promoted inflammation [23]. We found the induction of IL8 from each *H. pylori* was dependent on the dose and virulence of each *H. pylori*. The *H. pylori* strain ATCC 49503-infected 2-D gastric cancer cells showed rapid growth and the highest expression of HDGF, NF-*k*B, and IL8 [12], while both the moderately-toxic strain HpHtrA171/nonL and non-CagL Y58/E59 *H. pylori*-infected organoids and the non-cytotoxic strain *H. pylori* CagLm Y58D/59K-infected organoids showed the lowest invasion activities and growth of the organoids [18,22]. However, they also did not affect cell viabilities in the infected organoids.

Here we focused on HDGF and TNFα, which might affect the cell viability (Figure 2C,D). To clarify the role of HDGF and TNFα secreted from *H. pylori*, we prepared recombinant proteins of HDGF and TNFα and added them into the organoid cultures, and the cytotoxicity of gastric cancer organoids. Surprisingly, HDGF and TNFα decreased the cell viability of the cancer organoids.

### 2.3. Expansion of Human Gastric Cancer Organoids Infected by H. pylori and Effects of Recombinant HDGF and TNFα on Their Viabilities and Invasion Activities

Previous studies suggested that TNFα is possibly the upstream target of HDGF [12]. In this article, TNFα is a priming signal to increase *H. pylori*-induced TNFα/COX-2/PGF2 expression. Then, NF-*k*B signaling plays a crucial role in inflammation, and targets HDGF secretion [12].

We prepared recombinant proteins of HDGF and TNFα and purified them for use in the cytotoxic assays of treatment with *H. pylori.* In this system, we divided the organoids into 24 wells and infected *H. pylori* for 1 day followed by treatment with 10 or 100 ng/mL HGDF or 100 or 500 ng/mL TNFα for 7 days (Figure 3A,B). The cytotoxic assays demonstrated that only ATCC 49503 showed a significant increase in cell viability in the presence of HDGF at 100 ng/mL, but not other strains, such as Hp1033, HpCagLm Y58D/E59K, and HpHrtA171/nonL and nonCagL Y58/E59. At the high concentration of 500 ng/mL TNFα, the organoids infected by all *H. pylori* strains examined showed reduced cell viability (Figure 3B).

In the case of invasion assay, the results of HDGF and TNFα were variable depending on infection with different strains of *H. pylori.* In this assay, after one week of incubation, the organoids were dissociated with TrypLE™ Express Enzyme (Thermo Fisher Scientific, Waltgam, MA, USA) and cells were transferred into Transwell plates with the Matrigel on the surface, for cultivation for 6 days and then the growth and organoid invasion area were evaluated. Only ATCC 49503-infected organoid demonstrated enhanced invasion activity following incubation with HDGF (10 and 100 ng/mL). By contrast, the invasion activities of both Hp1033- and HpCagLm Y58D/E59K- infected organoids decreased after exposure by HDGF at 10 or 100 and 10 ng/mL, respectively. In the case of TNFα, the invasion activities of ATCC 49503- and Hp1033-infected organoids significantly decreased at 100 and 500 ng/mL. HpCagLm Y58D/E59K-infected organoid showed a similar decrease in invasion activity after exposure of TNFα at 100 ng/mL. HpHrtA17/nonL and nonCagL Y58/E59-infected organoids showed a slight increase in invasion activity at 500 ng/mL TNFα. These data demonstrated that the effects of *H. pylori* strain on cell viability and invasion activity after exposure of HGDF and TNFα were variable, dependent on *H. pylori* strain and the concentrations of HGDF and TNFα (Figure 3A–D). Chu et al. reported that ATCC 49503 infected 2-D gastric cancer cells showed rapid growth and high expression of HDGF and TNFα might be present upstream of the cascade of *H. pylori*-mediated cascade [12]. However, this was not consistent with our data using the recombinant HDGF and TNFα in human gastric cancer 3-D-organoid system.

To further confirm this issue, CagA-GFP-labeled *H. pylori* Hp0547-infected organoid clones were used for cell viabilities and invasion activities in the presence of various recombinant HDGF and TNFα proteins (Figure 4). As shown in Figure 4A, CagA-GFP-labeled *H. pylori* Hp0547-infected organoid clones were examined for cell viabilities in the presence of various recombinant HDGF and TNFα proteins. In all cases, the recombinant HDGF and TNFα were inhibitory to the growth of 3-D cancer organoids (Figure 4B).

The invasion activity of CagA-GFP-labeled Hp0547-infected organoid showed a 1.2-fold increase. The addition of recombinant HDGF did not significantly affect the invasion activity of *H. pylori*-infected clones (Figure 4C,D). In addition, the recombinant TNFα reduced this invasion activity considerably compared with the invasion potency of clones without TNFα (Figure 4C,D).

Only the *H. pylori* ATCC 49503 strain demonstrated increased invasion of gastric cancer organoids in the presence of HDGF, which observation is consistent with our previous reports [12], but we did not detect the same results in other *H. pylori* strains. We used the same strain of *H. pyori* on 2-D and 3-D experiments (Figure 3D). Further studies are required to examine the roles of both HDGF and TNFα in organoid viability and invasion capacity for gastric cancer progression and to identify the signaling pathways in *H. pylori* infection.

### 2.4. Survey of High and Low HDGF and TNFα in H. pylori-Infected Patients

To survey the correlation between the expression of HDGF and TNFα with the clinical outcome, we surveyed the mRNA expression of these genes in 231 patients. As a control, each tumor tissue was paired with neighboring healthy stomach tissues and analyzed based on the Gene Expression Profile Interactive Analysis (GEPPIA2) portal (http://gepia2.cancer-pku.cn/#survival, accessed on 10 May 2019) [24] (Figure 5). To characterize the clinical patterns and evaluate the prognostic value of HDGF and TNFα in *H. pylori*-infected gastric tumors, the data from 231 patients with adrenocortical carcinoma (ACC) or stomach adenocarcinoma (STAD) were normalized by the expression of CagA expression. The baseline characteristics of these tissues were divided into “low” and “high” expression groups for these genes using the survival receiver operating characteristic curve analysis.

The survival curve analysis of patients with ACC and STAD showed a significantly longer survival time after *H. pylori* infection in patients with high TNFα expression from the early stages than from the later stages (Figure 5B). By contrast, the survival time was observed to be similar after *H. pylori* infection in patients with higher or lower HDGF expression (Figure 5A). These data imply that the actions of HDGF and TNFα might tend to play independent roles in the generation of gastric cancer.

## 3. Discussion

We have used GFP-labeled *H. pylori* Hp0547 to trace the infection stage to search the entering pathways. In general, it has been reported that *H. pylori* directly colonizes into the surface of gastric cancer stem cells and adheres to gastric mucosa epithelial cells, and then attaches to superficial cell pits [25,26]. We found that the GFP signal of GFP-labeled *H. pylori* 0547 was present in the organoids for at least one week after injection. It might be useful to trace GFP-labeled *H. pylori* in gastric cancer for therapeutic use (Figure 1). We also compared cell cytotoxicity and growth of three different strains with a different toxicity for oncogenes. However, the cytotoxicity of HCM-BROD-0045-C16 was not affected by these strains of *H. pylori* (Figure 2A,B). We expected a range of cytotoxic activities of the gastric cancer organoids infected with each *H. pylori*, from strong infectivity to weak infectivity. However, we found no difference among the strains. The exposure of recombinant proteins of HDGF and TNFα to HCM-BROD-0045-C16 organoids reduced the cell viability of 3-D organoid cells (Figure 2C,D). Interestingly, the cell viability of HCM-BROD-0045-C16 organoids was significantly decreased in a dose-dependent manner. In the case of infection by *H. pylori* ATCC 40593, recombinant HDGF at high dose (100 ng/mL) increased the cell viability of 3-D organoids (Figure 3A,B). Other strains such as Hp1033, HpCagLm Y58D/59K, and HpHtrA171/nonL and non-CagL Y58/E59 exhibited the cell cytotoxic activities of TNFα (Figure 3A,B).

In the case of assay of invasion abilities of HCM-BROD-0045-C16 organoids, we compared the effects of HDGF with those of TNFα. In terms of inhibition of cell viability, we observed contradictory effects of HDGF and TNFα on 3-D human gastric cancer organoids (Figure 4A,B) that were infected with CagA-GFP-Hp0549. In the case of the invasion assay, recombinant HFDGF did not affect the growth of CagA-GFP-Hp0549, but recombinant TNFα inhibited the invasion activities (Figure 4C,D). Using this organoid system, we provide evidence against the signaling pathways of TNFα and HDGF to develop gastric cancer.

We used gastric adenocarcinoma-derived organoids from ATCC as the source of gastric cancer because this population can form tumors in xenotransplantation tests, indicating the presence of stem cells in adult human gastric tissue. These cells can be differentiated into specific lineages of the stomach (unpublished data). The stem cell marker LGR5 is a marker of antrum long-lived stem cells [27]. LGR5+ stem cells in the base of the antrum glands drive the repopulation of the glands by giving rise to highly proliferative progenitor cells. These progenitor cells, located in the mid-glandular compartment, rapidly divide, and differentiate into all epithelial lineages [25]. The amino acid polymorphism of *H. pylori* CagL-like Y58E59 was associated with a higher risk of gastric cancer and may regulate a corpus shift of gastric integrin α5β1, leading to severe corpus gastritis during gastric carcinogenesis [18,22]. We compared the toxicity of these mutants and found that the order of the toxicity of these three strains was consistent with the invasion activities of mutants of CagL. It is plausible that invasion activity might be a simple method to measure cancer potency instead of xenotransplantation or gastric implantation or kidney replacement for stomach surgery by endoscopy.

In addition, it was reported that TNFα growth factor secreted from *H. pylori* infected gastric cancer cells was the upstream target factor of HDGF [12]. Whether or not the recombinant proteins of HDGF and TNFα enhanced the invasion activities of 3-D organoids were evaluated. We observed the inhibitory activity of cell growth of the organoids in response to TNFα and HDGF, and inhibition of the invasion activity in response to TNFα but no change in the case of HDGF, suggesting that these signaling pathways are different. The human clinical data of ACC and STAG showed that HDGF and TNFα acted inversely to produce cancer phenotypes. High expression of TNFα might be associated with the higher survival curves (Figure 5). However, in the case of HDGF, a difference in survival between the higher and lower groups was not detected (Figure 5). We speculate that patients with higher TNFα are able to kill the ACC and STAG cancer stem cells selectively, indicating that the surviving cells might persist longer in healthy patients.

These results indicate that the effects of 3-D organoids and 2-D cells might provoke a different outcome for gastric cancer. To use the organoids for cancer therapy, several points remain to be solved. At present, the need to test the invasiveness or metastatic potential of cancer organoids in mouse xenograft models is without robust alternatives [28]. Research into finding suitable replacements for such models should be fostered in the future. In fact, *H. pylori*-infected organoids have difficulty growing in the xenograft to form the tumors in our model system. Schlaermann and colleagues developed a monolayer culture from similar 3-D gastric organoids [29]. In contrast to observations in 2-D cells, the effect of HDGF and TNFα using *H. pylori* 3-D organoids has not been demonstrated. Development of a therapeutic use for 3-D organoids remains a challenge and will depend on new techniques to adapt the organoids for preclinical use such as drug screening and the growth factor identification for supporting organoids beside Matrigel. The organoids infected by *H. pylori* are sometimes difficult to grow to the cancerous stage and it is difficult to generate the tumors from xenografts. In addition, tumor formation might take a long time. Instead, we used the invasion assays to replace the xenograft assays because it is a rapid assay of tumorigenesis focused upon the invasion ability.

This invasion assay is efficient and accurate for monitoring cancer. We confirmed the results of recombinant HDGF and TNFα using 3-D organoids, but it took about 2–3 months by xenotransplantation assays, and the tumor size was always very small. Thus, we recommend the invasion assay instead of xenografts assays to measure tumorigenesis. In this paper, we found the invasion assay to be the most efficient assay to examine potential tumorigenesis.

## 4. Materials and Methods

### 4.1. Helicobacter pylori Strains, Culture, and Microinjection

The CagA-*H. pylori* strain (ATCC 49503, American Tissue Type Collection, Manassas, VA, USA) and three different extents of the toxin, namely the CagL mutant strain Hp1033 (highest toxic activity) [22], HpHtrA171/nonL and nonCagL Y58/E59 GC strain (lowest toxic activity), and HpCagLm Y58D/E59K (nontoxic ability), were obtained from B-S. Sheu [18,22]. Triplicase soy agar with 5% sheep blood agar plates (BD Biosciences, Bedford, MA, USA) was for cultivation of *H. pylori* bacteria. *H. pylori* were grown in Brucella broth (BD Biosciences) supplemented with 10% fetal bovine serum (FBS) for 16 h at 37 °C with 5% CO_2_ to amplify and activate the virulence. CagA-enhanced GFP-labeled *H. pylori* Hp0547 was a gift from Dr. Ping-Ning Hsu, National Taiwan University, Taiwan [30] and microinjected into the human gastric cancer organoids as described previously [30]. In brief, for infection, the organoids were seeded in 50 mL Matrigel in 4-well multi-dishes (Thermo Scientific, Waltham, MA, USA). Antibiotic-free medium Dulbecco’s Modified Eagle Medium (DMEM)/Ham’s Nutrient Mixture F-12 (Invitrogen, Waltham, MA, USA) was refreshed every 2 days, with three medium changes at least before infection to allow exclusion of antibiotics from the culture. Organoids were microinjected on day 10 after seeding with a multiplicity of infection (MOI) of 100 or 200 unless it was stated otherwise. For calculation of MOI, organoids were disrupted into single cells by Ethylenediaminetetraacetic acid and cells were counted (approximately 4000 cells/organoid). To achieve a MOI of 100, bacteria were diluted in advanced DMEM/F12 at 10^9^/mL and then organoids were injected with approximately 0.4 mL of *H. pylori* suspension using a micromanipulator and microinjector (M-152 and IM-5B; Narishige, Tokyo, Japan). Epithelial cells were infected by *H. pylori* at a MOI of 100 or 200 at different time points, following standard protocols [31].

### 4.2. Organoid Preparation and Preparation of Recombinant HDGF and TNFα Proteins

Gastric organoids were prepared from human stomach organoid of adenocarcinoma, HCM-BROD-0045-C16, purchased from American Type Culture Collection (PDM-116™; ATCC, Manassas, VA, USA) and cultured using its recommended protocol with modification [8]. The gastric organoids were infected with the indicated H. pylori strain for 1 day. The viability of organoids was determined using CellTiter-Glo*^®^* 3D Cell Viability Assay (Promega, Madison, WI, USA; G9683) as described by its recommended protocol. The gastric organoids morphologies were recorded and scanned by Cell3iMager neo scanner (CC-3000; SCREEN Holdings Co., Ltd., Kyoto, Japan). Recombinant HDGF and TNFα proteins were obtained from Dr. MH Tai (Ref. [12]) and PeproTech (Granbury, NJ, USA).

### 4.3. Bioreactor Cultivation

This method was adapted from the simple bioreactor-based method described by Peplowski et al. [32] with a slight modification for gastric organoids. The gastric organoids were prepared for culture in a 5 mL or 30 mL disposable bioreactor (Able, #BWV-S005A, S03A) on a magnetic stirring base plate, and resembled embryoid body structures naturally formed in the bioreactor for 3-D organoid suspension culture by using ABLE Biott Bioreactor System (REPROCELL Inc., Yokohama, Japan).

### 4.4. Measurement of Viable Cells by 3-D Cell Viability Assay

The CellTiter-Glo 3-D Cell Viability Assay is performed according to the manufacturer’s instructions to determine the number of viable cells in 3-D cell culture based on quantitation of the ATP present, which is a marker of the presence of metabolically active cells (Promega). The CellTiter-Glo 3-D Cell Viability Assay is formulated with robust lytic capacity and is designed for use with microtissues produced in 3-D cell culture.

### 4.5. Invasion Assay

Organoids were treated by TryLE^TM^ Express Enzyme12694013 (ThermoFisher Scientific Inc., Waltham, MA, USA) to prepare the spheroplasts-like cells. Then, these gastric organoids were transferred to individual 8.0-μm Transwell plates (Costar; #3422, Corning Incorporated, Corning, NY, USA) and coated with Matrigel (Corning Incorporated, Corning, NY, USA; 1 mg/mL) without serum. The Transwell plates were then put on a plate containing indicated organoids in culture medium for 7 days by the invasion assay [33,34]. The invaded cells on the lower surface of membrane were fixed, stained, and counted under a microscope according to the manufacturer’s instructions.

### 4.6. Cancer Genome Atlas/GEPIA2 Portal Analysis

Large-scale cancer genomics projects such as GEPIA2 (http://gepia2.cancer-pku.cn/#survival, accessed on 10 May 2019) data [24] were accessed, and we surveyed gene mutation maps of HDGF and TNFα proteins. Finally, 231 and 230 samples from 125 and 83 studies, respectively, were summarized in GEPIA2 for Cancer Genomics in Adrenocortical and Stomach adenocarcinoma tumors.

### 4.7. Statistics

Data are presented as the mean ± standard error of the mean. Statistical comparisons between experimental conditions were conducted using GraphPad Prism 5.0 (GraphPad Software, San Diego, CA, USA). For multiple comparisons, a one-way analysis of variance (ANOVA) and then Tukey’s test were performed using GraphPad Prism 7.0 (San Diego, Ca, USA). An unpaired, two-tailed Student’s *t*-test was used to compare the control and treatments. The Mann–Whitney nonparametric median statistical test was used for analysis of cell areas. All differences were designated as statistically significant at *p* < 0.05.

## 5. Conclusions

The signaling of *H. pylori* infection on human gastric cancer organoids has not been well studied. Here, we provide evidence of different signaling for cancer progression and invasion of human gastric organoids in response to HDGF and TNFα during infection by *H. pylori*. While TNFα is defined as the upstream partner of HDGF in 2-D gastric cancer cells, in 3-D organoids in culture, HDGF and TNFα are independent signals for development of *H. pylori*-infected gastric cancer. Recombinant HDGF and TNFα inhibited the growth and invasion activities in 3-D-gastric cancer organoids.

## Figures and Tables

**Figure 1 ijms-24-06567-f001:**
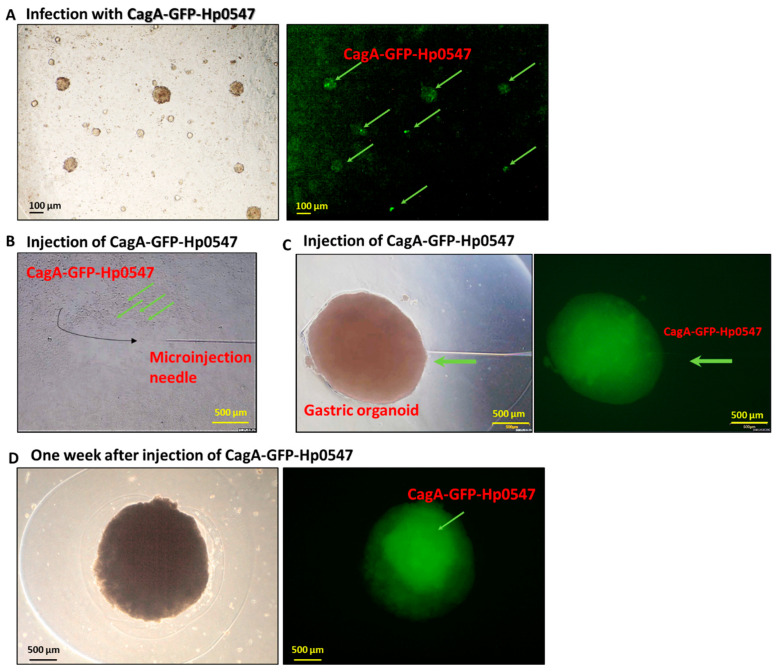
Comparative infection and microinjection of *H. pylori* into human gastric organoids and traced by green fluorescence protein signaling. (**A**) CagA-green fluorescence protein (GFP)-labeled *H. pylori* Hp0547 (CagA-GFP-Hp0547)-infected organoids were cultured for 24 h at a multiplicity of infection (MOI) of 200 and their signals were traced by fluorescence microscopy. Brightfield image (left panel) and their CagA-GFP expressed *H. pylori* Hp0547 image (right panel) are shown. (**B**) CagA-GFP-labeled Hp0547 was microinjected as described in Materials and Methods. The microinjection needle is also indicated. (**C**) Brightfield image of CagA-GFP-labeled Hp0547-infected organoids and their detection of GFP signal by CagA-GFP-Hp0549. (**D**) Brightfield image and GFP detection of CagA-GFP-Hp0549 after culture for 1 week. Scale bars indicated in each panel. Arrow indicates GFP signal from CagA-GFP Hp0549.

**Figure 2 ijms-24-06567-f002:**
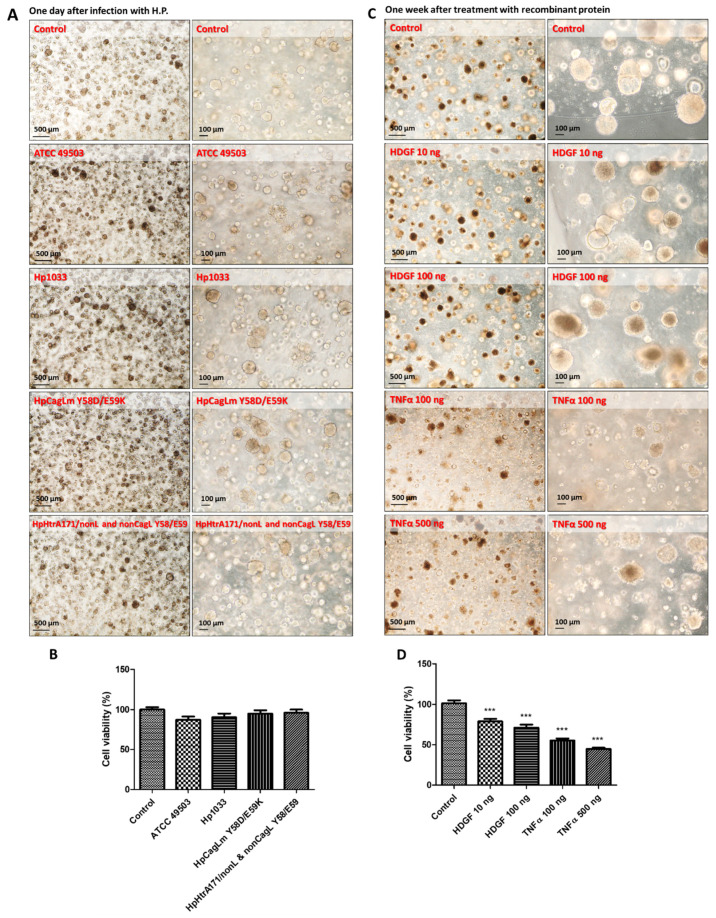
Cultivation of organoids after infection with various *H. pylori* species and examination of their viabilities. (**A**) The morphologies and growth of the gastric organoid HCM-BROD-0045-C16 for 1 day after infection with various *H. pylori* such as ATCC 49503, Hp1033, HpCagLm Y58D/E59K, and HpHrtA171/nonL and nonCagL Y58/E59 strains. (**B**) Cell viability of human gastric cancer organoids with infection by various *H. pylori* as described above. (**C**) The morphologies and growth of the gastric organoids HCM-BROD-0045-C16 for 7 days in the presence of indicated doses of HDGF and TNFα after infection with various *H. pylori* such as ATCC 49503, Hp1033, HpCagLm Y58D/E59K, and HpHrtA171/nonL and nonCagL Y58/E59 strains. (**D**) Cell viability of human gastric cancer organoids cultured for one week in the presence of indicated doses of recombinant HDGF or TNFα. The recombinant proteins were homogenous (ref. [12] or 300-01A, PeproTech, Cranbury, NJ, USA). *** indicates *p* < 0.005 (*n* = 6).

**Figure 3 ijms-24-06567-f003:**
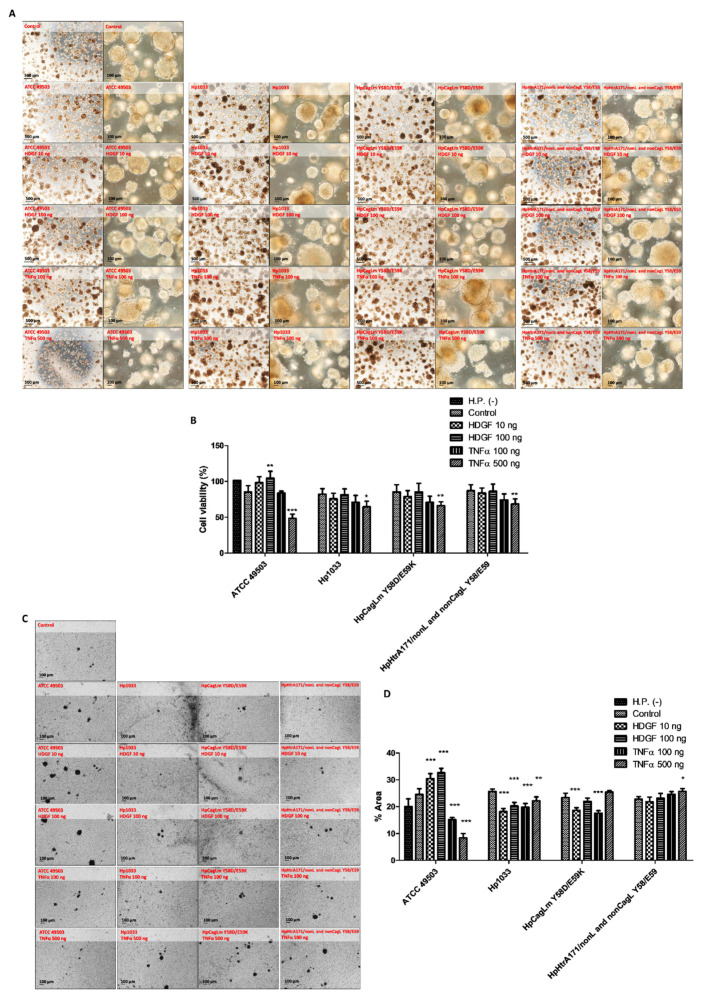
Cell viabilities and the area of the organoids after cultivation of organoids infected by various *H. pylori* species. (**A**,**B**) The morphologies and growth of the gastric organoids HCM-BROD-0045-C16 for 7 days after infection with various *H. pylori* such as ATCC 49503, Hp1033, HpCagLm Y58D/E59K, and Hp HrtA171/nonL and nonCagL Y58/E59 strains in the presence and absence of indicted doses of recombinant HDGF and TNFα. (**C**,**D**) The invasion activities of the organoids of HCM-BROD-0045-C16 for 7 days after infection with various *H. pylori* such as ATCC 49503, Hp1033, HpCagLm Y58D/E59K, and HpHrtA171/nonL and nonCagL Y58/E59 strains in the presence and absence of indicted doses of recombinant HDGF and TNFα. *, **, and *** indicate *p* < 0.05, *p* < 0.01, and *p* < 0.005 (*n* = 4), respectively.

**Figure 4 ijms-24-06567-f004:**
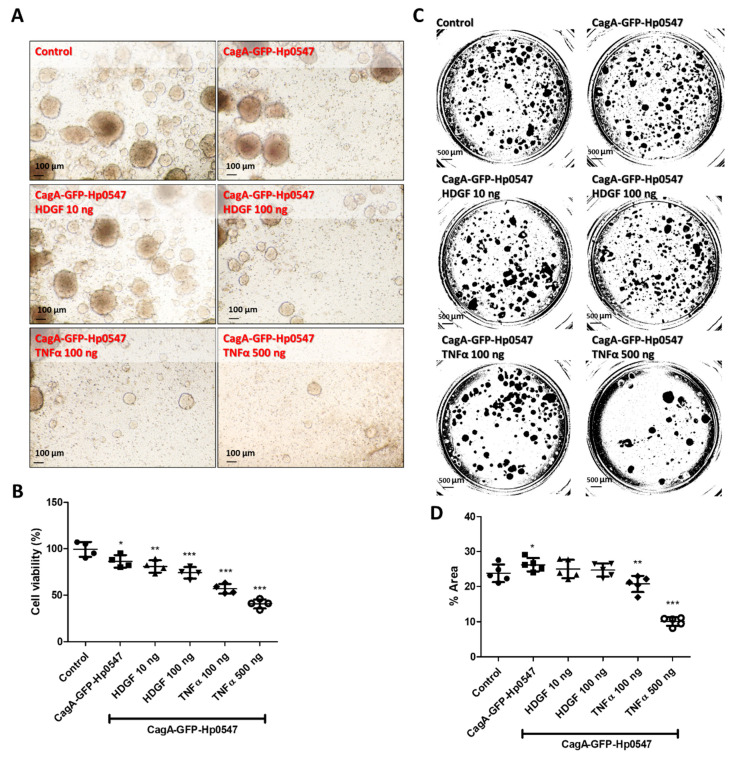
Cell morphologies and viabilities of the CagA-GFP-Hp 0547 labeled organoids after cultivation of organoids in the presence of various dose of recombinant HDGF and TNFα. (**A**,**B**) The morphologies and viabilities of the gastric organoids HCM-BROD-0045-C16 for 7 days after infection with CagA-GFP-Hp0547 in the presence and absence of indicted doses of recombinant HDGF and TNFα (*n* = 4). (**C**,**D**) The invasion activities of the gastric organoids infected by CagA-GFP- Hp0547 for 7 days in the presence and absence of indicted doses of recombinant HDGF and TNFα. *, **, and *** indicate *p* < 0.05, *p* < 0.01, and *p* < 0.005 (*n* = 5), respectively.

**Figure 5 ijms-24-06567-f005:**
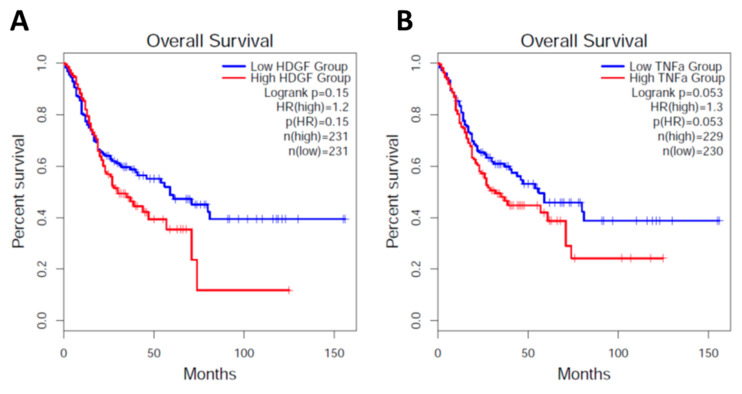
Survival curve of patients with high or lower expressed HDGF or TNFα. A total of 231 patients were surveyed using GEPIA2 (http://gepia2.cancer-pku.cn/#survival, accessed on 10 May 2019) as described in the Methods section; (**A**,**B**) are non-normalized by the CagA gene; (**C**,**D**) are normalized by the CagA gene.

## Data Availability

The data that support the findings of this study are available from the corresponding author upon reasonable request.

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
