# Peer review of "Independent Signaling of Hepatoma Derived Growth Factor and Tumor Necrosis Factor-Alpha in Human Gastric Cancer Organoids Infected by Helicobacter pylori"

_ijms, 2023, doi:10.3390/ijms24076567_

Round 1
Author Response
Reviewer 1:
Major concerns: Authors have evaluated majorly the outcome of the treatment and/or infection with H. pylori in terms of cell viability and invasiveness but not the signaling per se.
Answer: Thank you for your kind comment. We agree with the reviewer’s criticism. We have examined the effects of the recombinant proteins, HDGF and TNFa, on invasion activity and cell viability. Thus, we avoid the use of the term “signaling” to prevent any confusion because we used recombinant proteins. In terms of the signaling of HDGF and TNFa using 2D cancer cells in vivo, one of our coauthors has reported on this previously (TH Chu et al. Oncogene 38, 6461-6477, 2019). Here, we used recombinant proteins to confirm this observation in our artificial system of H. pylori-infected 3-D organoids. Thus, we have revised the text to avoid the confusion.
Line 107 – question: antrum-derived organoids and fundus- derived organoids were mentioned, however it is not clear that the data were generated from antrum or fundus derived organoids?
Answer: Thank you for your question. We used the human gastric adenocarcinoma primary cells derived from total portions of antrum and fundus. We have revised the text (Line 109).
Line 156 - The data Fig 2B represents cell viability, so then why authors mentioned as “we did not compare the…”
Answer: Thank you for your kind comment. As suggested by the reviewer, this is a typographical error. Thus, we deleted this sentence.
Line 161 – Authors mentioned about invasion ability, but the corresponding data is not presented. Please clarify this.
Answer: Thank you for your kind comment. This sentence on cell invasion was a typographical error. Thus, these words have been deleted.
Line 262 – “Figure 1” however the figure 1 does not demonstrate the localization of H. pylori colonies in gastric glands! Rather the GFP images only indicates that the bacteria fluoresce in organoid lumen.
Answer: Thank you for your criticism. We agree with the reviewer and have deleted these sentences. We described the images to emphasize that GFP images from GFP-H. pylori could be obtained in text and Figure legends.
Other required change
Line 54 – please remove “affecting which” and change to “which infected”
Answer: Thank you for your correction. We have revised it.
Line 55 – “H. pylori infection is associated with…” remove “used to be
Answer: Thank you for your correction. We have removed the phrase.
Line 56 – Please modify the sentence “Chronic infection lead to gradual progression of gastric atrophy…”
Answer: Thank you for your correction. We have revised the sentence
Line 58 – “H. pylori infection led to emergence of tumorigenic subpopulation cells that can…”
Answer: Thank you for your correction. We have revised the text.
Line 64 – “Even then, many questions remain unaddressed regarding the …
Answer: Thank you for your correction. We have revised the text.
Line 69 – remove “of” change to “leading to inflammation.”
Answer: Thank you for your correction. We have revised the text.
Line 85 – “TNF was identified/reported as an upstream partner of HDGF in gastric cancer cells in 2D culture”. Please provide citation.
Answer: Thank you for your correction. We have added the cited reference (TH Chu et al. Oncogene 38, 6461-6477, 2019).
Line 87 – “However, in this study we report that in three-dimensional…”…..“TNFa are involved in independent signals in..”
Answer: Thank you for your correction. We have revised the text.
Line 88 – “We also discuss this difference in growth factor signaling in …”
Answer: Thank you for your correction. We have revised the discussion.
Line 93 – “the antrum by following…”
Answer: Thank you for your correction. We have revised the text.
Line 94 – “HD1033” – Please use a standard method to mention H. pylori strains and use it uniformly throughout the manuscript. Also, HP 49503 and other strains in the manuscript.
Answer: Thank you for your kind advice. HD1033 is a typographical error and should be Hp1033. We cited the references (refs. 18 and 22) for Hp1033. HP49503 is the strain from ATCC and named ATCC 49503 as officially indicated. The nomenclature of other strains was also revised correctly.
Line 98 – “GFP expressed by H. pylori..”
Answer: Thank you for your correction. We have revised the text.
Line 113 – change “signaling” to signals Line 115 – “signals.”
Answer: Thank you for your correction. We have revised the text.
Line 118 – please cancel – “signaling of H. pylori” or change to “GFP signal by GFP H.P.”
Answer: Thank you for your correction. We have revised it.
Line 130 – “commonly exist with
Answer: Thank you for your correction. We have revised the text.

Reviewer 2 Report
This manuscript describes the role of HDGF and TNF alpha as signaling molecules in gastric cancer organoids infected with H.pylori. Below are some of my comments for this manuscript.
1) For the CagA gene, please write the full form where it is first introduced and then include the abbreviated form in parantheses. You can't mentione the abbreviated form in the beginning without mentioning the full name of the protein.
2) I'm not quite sure of the reason for adding lines 123-135. It does not add anything significant to the overall paper and to the result section 2.2.
3) The authors noticed that HDGF derived from H. Pylori when infected did not decrease cell viability whereas injection of rHDGF decreased cell viability. Was the recombinant protein the same as the one that was endogenously produced? Or do the endogenous ones have some PTM which may not be present in the recombinant HDGF.
4) Line 66-67. The authors have mentioned that HDGF and TNF alpha are secreted by H.pylori but in lines 66-67 it seems as though they are produced by the cancer organoids. This is not clear and has to be mentioned explicitly. It still does not come across as very clear in the entire anuscript.
5) Can you show the IL8 and NFkB data too? I think they are important in the context of this study.
6) Lines 192-199. It seems that the authors re-seeded the organoids by dissociating them now making them a 2D morphology and it might change the way the H.pylori interacts with the cells.
7) Fig 4. I'm unclear whether the data in all of the figures corresponds to which type of H.pylori strain.
8) This manuscript would greatly benefit from English editing.
4)
Author Response
Reviewer 2
This manuscript describes the role of HDGF and TNF alpha as signaling molecules in gastric cancer organoids infected with H.pylori. Below are some of my comments for this manuscript.
- For the CagA gene, please write the full form where it is first introduced and then include the abbreviated form in parentheses. You can't mention the abbreviated form in the beginning without mentioning the full name of the protein.
Answer: Thank you for your helpful comment. As suggested, we have described the full name of the cytotoxin-associated gene A (CagA) and provided the abbreviated form in parentheses.
- I'm not quite sure of the reason for adding lines 123-135. It does not add anything significant to the overall paper and to the result section 2.2.
Answer: Thank you for your constructive comment. We agree with your point. Thus, we have deleted these lines.
- The authors noticed that HDGF derived from H. Pylori when infected did not decrease cell viability whereas injection of rHDGF decreased cell viability. Was the recombinant protein the same as the one that was endogenously produced? Or do the endogenous ones have some PTM which may not be present in the recombinant HDGF.
Answer: Thank you for your comment. Recombinant HDGF and recombinant TNFa are produced in E. coli, and the backbones of the protein sequences are identical in structure to natural human HDGF and TNFa except for post-modifications such as ubiquitination and phosphorylation. In terms of the growth, invasion, anchorage-independent growth and tumorigenicity, and signaling of AKT/HIF-1alpha/NF-kB and nucleolin-dependent PI3K/AKT cascade in oral cancer and liver cancer and various immunological responses, exogenous recombinant proteins can be functional as reported (TH Chiu et al., Oncogene 31, 3280-3829, 2019; Y-W Lin et al., BMC Cancer 11, 1083, 2019; Sc Chen et al., Oncotarget 6, 16253-16270, 2015; MS Shutova et al., iScience 26, 1061195, 2023; CD Liu et al., EMBO J, 42, e111614, 2023). These recombinant proteins of HDGF and TNFa showed differently on the cancer related functions such as cytotoxicity and invasion of H. Pylori infected cancer 3D organoids. These data indirectly support that HDGF and TNFa function differently in vitro. We mentioned this in Discussion section.
- Line 66-67. The authors have mentioned that HDGF and TNF alpha are secreted by H.pylori but in lines 66-67 it seems as though they are produced by the cancer organoids. This is not clear and has to be mentioned explicitly. It still does not come across as very clear in the entire manuscript.
Answer: Thank you for your criticism. According to our previous reports using 2-D cancer cells (TH Chiu et al., Oncogene 31, 3280-3829, 2019), we know that HDGF and TNFa can be produced from cancer cells after infection with H. pylori. Other reports also show that both factors as well as TNFa-interacting protein are secreted in human mesenchymal stem cells (Cancers, 10, 479, 2018; Pathog Dis., 80, dtac025, 2022; L Med Life, 15, 4-6, 2022; J Biol Chem 289, 27776-27793, 2014; Phytomedicine 63, 152968, 2019; Cell Rep., 42, 112005, 2023). In the case of the organoids, there are no reports to show direct secretion at this moment. Here, we did not have direct evidence that the organoids can secrete HDGF and TNFa because we did not use the ELISA assay of the culture supernatant of the 3-D organoids. Specifically, no reports showed the secretion of HDGF and TNFa from the gastric cancer 3-D organoids infected by H. pylori. To avoid confusion, we deleted this sentence from the text.
- Can you show the IL8 and NFkB data too? I think they are important in the context of this study.
Answer: Thank you for your enthusiastic comments. In the sentence from lines 228-232 in the original text, some words were missing because of typographical errors. The complete sentences are below.
In the case of 3-D gastric cancer organoids, not only the invasion activities (Figure 2B) but also the production of IL8 and NFkB expression was unaffected (in various H. pylori-infected organoids. However, their invasion activity and expression are significantly different in the 3-D organoids treated) by the recombinant HDGF and TNFa in vitro (data not shown).
These data of IL8 and NFkB p65 were generated by qPCR analysis to measure the RNA levels that are shown below. However, we need to study these issues in the future studies. Thus, in this paper, we chose not to show these data (data not shown) and deleted lines 228-232 in an original text.
The data on qPCR are shown below.

- Lines 192-199. It seems that the authors re-seeded the organoids by dissociating them now making them a 2D morphology and it might change the way the H. pylori interacts with the cells.
Answer: Thank you for your kind criticism. We agree with the reviewer’s opinion. The problem is a technical issue. The 3-D organoids use Matrigel to generate the 3-D structure. In the case of the invasion assay, we used an apparatus to measure the invasion activity where the wells were coated with a different type of Matrigel. This is why we used the digestion to break the 3-D cluster of the organoids briefly before applying it to the invasion apparatus. The methods were described in the following references (Sato T, Stange DE, Ferrante M, et al.: Long-term expansion of epithelial organoids from human colon, adenoma, adenocarcinoma, and Barrett's epithelium. Gastroenterology. 2011 141: 1762–1772; De Angelis ML, Francescangeli F, Nicolazzo C, et al.: An organoid model of colorectal circulating tumor cells with stem cell features, hybrid EMT state and distinctive therapy response profile. Journal of experimental & clinical cancer research. 2022 41:86).
.
4.5. Invasion assay
Organoids were treated by TryLETM Express Enzyme12694013 (ThermoFisher Scientific Inc., Waltham, MA, USA) to prepare the spheroplasts-like cells. Then, these gastric organoids were transferred to individual 8.0-mm Transwell plates (Costar; #3422, Corning, NY, USA) and coated with Matrigel (Corning, NY, USA.;1mg/mL) without serum. The Transwell plates were then put on a plate containing indicated organoids in culture medium for 7 days by the invasion assay [33, 34]. The invaded cells on the lower surface of membrane were fixed, stained, and counted under a microscope according to the manufacturer’s instructions.
- Fig 4. I'm unclear whether the data in all of the figures corresponds to which type of H.pylori strain.
Answer: Thank you for your kind advice. We have revised the figures to make sure they describe the correct names of the H. pylori strains.
CagA-GFP-Hp0547
ATCC 49503,
Hp1033,
HpCagLm Y58D/E59K,
HpHtrA171/nonL and nonCagL Y58/E59
- This manuscript would greatly benefit from English editing.
Answer: Thank you for your kind suggestion. The manuscript has been extensively revised by the Online English editing company.

Reviewer 3 Report
No further comments
Author Response
Reviewer 3
No further comments.

Round 2
Reviewer 2 Report
The authors have satisfactorily addressed the comments made earlier in my review.